# Delays in generalization match delayed changes in representational geometry

**Xingyu Zheng,**[*]   **Kyle Daruwalla,   Ari S. Benjamin,   David Klindt**
Cold Spring Harbor Laboratory
Cold Spring Harbor, NY 11724
{xinzheng, daruwal, benjami, klindt}@cshl.edu

## Abstract

Delayed generalization, also known as "grokking", has emerged as a well-replicated phenomenon in overparameterized neural networks. Recent theoretical works associated grokking with the transition from lazy to rich learning regime, measured as the change in the Neural Tangent Kernel (NTK) from its initial state. Here, we present an empirical study on image classification tasks. Surprisingly, we demonstrate that the NTK deviates from its initial state significantly before the onset of grokking, i.e., before test performance increases, suggesting that rich learning does occur before generalization. To explain this difference, we instead look at the representational geometry of the network, and find that grokking coincides in time with a rapid increase in manifold capacity and improved effective geometry metrics. Notably, this sharp transition is absent when generalization is not delayed. Our findings on real data show that lazy and rich training regimes can become decoupled from sudden generalization. In contrast, changes in representational geometry remain tightly linked and may therefore better explain grokking dynamics.

## 1 Introduction

Neural networks generalize to unseen data despite having capacity to overfit their training data. This phenomenon underlies much of modern machine learning, but our knowledge about it is still limited [16, 34]. One window that can help study generalization is "grokking" in which models generalize long after over-fitting their training set [27]. Grokking has been found in many machine learning tasks [19] and can be induced in many training paradigms [15, 31], suggesting it is a general phenomenon that may allow more detailed insights into the performance of neural networks [13].

One theoretical explanation of grokking is that it corresponds to a transition from the "lazy" linearized regime to the "rich" feature-learning regime [15, 20]. Lazy learning refers to a regime where the network function remains close to its initialization and the weights change minimally [4]. Such a network can be approximated as a kernel machine, where the learned function is a linear combination of kernel functions centered at the training points [14]. This kernel function, known as the neural tangent kernel (NTK) $\mathcal{K}$, is derived from the dynamics of gradient flow in the linearized network. It measures the similarity between inputs, and the learned function relies on the similarity between the test input and the training examples to make predictions. Prior to grokking, networks have been reported to exhibit a few key characteristics reminiscent of the lazy learning regime: exponential convergence of the network to perfect training performance [27], large test errors, and a correspondence with weight initializations with large weight norms [19, 25]. Indeed, Lyu et al. [20] demonstrated that in scenarios with large weight norms and weight decay, there is a sharp transition in the network's behavior, exhibiting different implicit biases in the early and late phases of learning. This transition, marked by "grokking", shifts the network from behaving more like a kernel regression model to

---

[*]Code available at https://github.com/cici-xingyu-zheng/rich-regime-geometry

acting as a maximum margin classifier. Kumar et al. [15] proposed that grokking corresponds to a transition from lazy to rich training dynamics and examined this phenomenon from two theoretical angles: 1) the alignment of the NTK to task features (in a theoretical setup), and 2) the scaling of model output as a rough proxy for the rate of lazy learning versus feature learning. By varying both the alignment and the scaling, they observed a continuous change in the timing of delayed test performance improvement. Kumar and colleagues' results suggested that grokking is primarily a function of two factors: the task-model alignment at initialization [3] and the rate of feature learning.

The transition from a kernel regime around initialization to a rich regime offers a straightforward explanation to the grokking phenomenon. However, it remains unclear whether this theoretical transition manifests in real-world task settings. Here, we aim to elucidate the nature of this transition and identify fundamental changes in the network as generalization occurs. We find that:

1. Contrary to our expectations, networks that grok in an image classification setting show large changes in their NTK *before* delayed generalization, revealing the complex nature of training dynamics in real-world task settings.

2. Introducing measures of the representational geometry, we see that delayed generalization instead coincides with a rapid improvement in object manifold capacity, signifying feature learning but by a different measure [6, 8].

3. Changes in network manifold capacity also allow us to identify a late stage over-fitting, where representations in over-trained network lose the ability to generalize.

## 2   Related work

**Empirical evidence for grokking as a feature-learning transition:**. We examine here a few studies that posit grokking as a transition from lazy to rich learning and claim to have provided empirical evidence for this hypothesis. Lyu et al. [20] first demonstrated in their theoretical setup that gradient flow can initially remain constrained within the kernel regime, with the model optimizing over a linearized representation before deviating from the NTK direction at initialization. To support this result, they utilized changes in parameter weight norm as a progress measure for grokking and they observed that these changes precede improvements in test performances in a sparse linear classification task and a matrix completion task. While this approach reveals a potential important causal mechanism for the emergence of grokking and offers a theoretical explanation, changes in weight norm alone are insufficient to fully characterize the transition in learning regime, as rotations preserve norms. Kumar et al. [15], who also conceptualized grokking as a transition from lazy to rich learning, employed output scaling to induce grokking in their empirical setup [4]. The output scaling parameter was utilized to modulate the network's gradient sensitivity with respect to changes in the loss function, thereby influencing the degree of laziness. While this approach successfully induces grokking in modular arithmetic task and MNIST image classification, it remains unclear whether the change in the network function actually follows lazy learning before grokking and

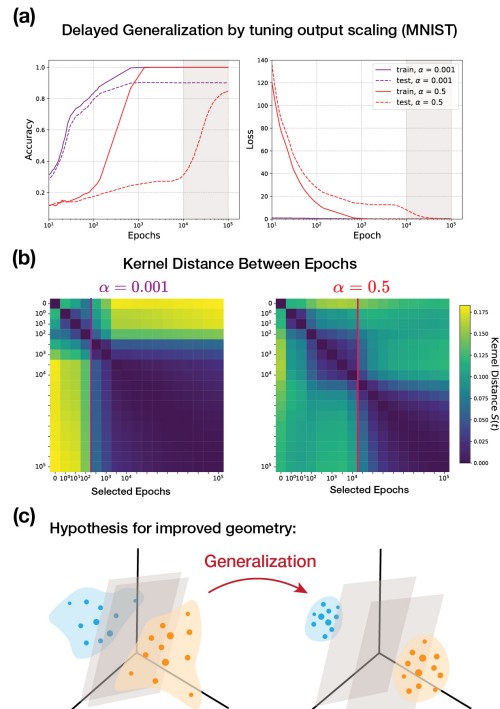

Figure 1: **Grokking might be associated with changes in manifold geometry.** **(a)** Reproducing induction of grokking in Kumar et al. [15] by tuning output scaling. **(b)** Pairwise kernel distance for selected epochs; colored line marks the rough onset of generalization. **(c)** Hypothesis of improved representation geometry associated with generalization. In object classification, it means the classes become more robustly separable.

rich learning after. A recent study by Mohamadi et al. [24] investigated why modular addition is fundamentally hard to generalize (thus grok). They measured the Frobenius norm of the difference between the time-evolving NTK and the NTK at initialization for the modular addition task. Based on the details provided, it is unclear whether changes in the kernel difference track training or test performance. Kunin et al. [16] present an unbalanced rich learning regime, where different layers of a network evolve with different effective learning rates. They showed that promote faster feature learning in earlier layers can accelerate grokking, i.e., reduce the delay in generalization on modular arithmetic tasks with transformers. While prior works have induced grokking for complex, high-dimensional distributed data (such as images), the transition from lazy to rich is more implied than demonstrated with metrics.

**Geometry of generalization and memorization:** Representational geometry has long been employed to elucidate encoding mechanisms in both artificial neural networks and biological brains. This approach is based on the principle that effective representations of inputs in neural networks should align closely with task-relevant features, thereby facilitating generalization performance [1, 2, 10]. The method we employ here, manifold capacity theory, was developed to formally connect the linear separability of object manifolds in a network with the underlying geometric properties using mean field theory techniques [5, 6]. The line of work has been used to understand various deep learning phenomena [8, 22, 29, 30, 33]. Intuitively, the manifold capacity $\alpha_C$ serves as measure of the critical load of linearly decodable information per unit neuron about object identity [18]. In a network with high manifold capacity, object classes should be better separated, and the shape/geometry of the manifolds that determines the value of the manifold capacity is also informative to understand the structure of representations. In particular, networks trained on classification tasks demonstrate improved manifold capacity compared to initialization, along with improved manifold geometry properties. Methods based on this theory have also been employed to look at other generalization problems such as few-shot learning [28] and memorization [30].

# 3 Experimental Setup

In the following, we propose a simple setup to induce grokking as well as to study learning dynamics and representation geometry in networks trained on image classification tasks. We focus on the task of image classification for its amenability to manifold geometry analysis and the intuitive interpretation it can offer for representation learning.

## 3.1 Dataset and Models

We adopt the same network as in [15] with the AdamW optimizer and MSE loss to mirror their results on MNIST first to see whether the implied lazy to rich transition holds. For the main results on the balanced EMNIST dataset, all models were trained using a 7-layer (hidden layers with 128, 512, and 128 neurons, and ReLU activation functions used between the layers). Models were either trained for 10,000 or 50,000 epochs, with AdamW optimizer (learning rate = 0.001) and the cross-entropy loss function.

We employ three manipulations to the training setup to control grokking, following the approach of Liu et al. [19] and Kumar et al. [15]. These are briefly described below (see Appendix Table 1 for details):

1. **Weight scaling and decay**: We increase the scale of the initial weights (default on Kaiming uniform initialization used in PyTorch) by a constant $\alpha' > 1$, and manipulate the initial weight norm subject to a fixed weight decay rate ($\gamma = 0.001$).

2. **Weight scaling and reduced training size**: We fix the initial weight scaling and weight decay rate, and select random subsets of the training data ($n = 1000, 2000, 5000$) to train the model.

3. **Output re-scaling**: We scale up the initial weight norms as above, and multiply the output of the network by a constant $\alpha$. $\alpha$ is used in keeping with the notations in [4] and [15] (not to be confused with the manifold capacity $\alpha_C$ that we introduce in the next section).

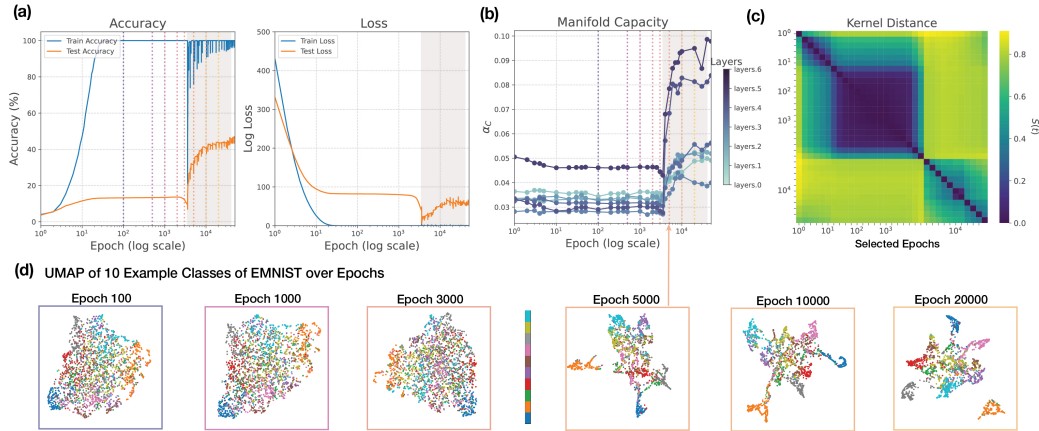

Figure 2: **Manifold capacity changes during grokking on the EMNIST dataset.** Grokking was induced in the small training size ($n = 1000$) with large weight norm and weight decay scenario. The "grokking" process is highlighted with shading, during which **(a)** the test accuracy increases, the test loss drops, and **(b)** the manifold capacity increases. **(b)** Manifold capacity starts to increase at the onset of grokking. **(c)** Pairwise kernel distance heatmap between selected epochs. **(d)** UMAP visualization at sampled epochs of the last layer activations (10 classes with $n = 10,000$ test samples).

## 4 Results

**Network function diverges from initialization before grokking.** We first reproduced the induced grokking for MNIST from [15, 19], by tuning either weight decay or output scaling (Fig. 1 a, Appendix Fig. 7). Consistent with the predictions of Lyu et al. [20], we observed that the weight norms of the networks initially exhibit minimal change prior to grokking, accompanied by a notable decrease coinciding with the networks' improved generalization (Appendix Fig. 8).

We then investigated whether grokking is coupled to changes in the Neural Tangent Kernel (NTK). The NTK describes the learning dynamics of a network $f(x; \theta_t)$ at training time $t$, $f : \mathbb{R}^{d_{\text{in}}} \times \mathbb{R}^{d_p} \to \mathbb{R}^{d_{\text{out}}}$ in terms of the tangent kernel, which has the following form for MSE loss and cross entropy loss:

$$\mathcal{K}_t(x, x') = \nabla_\theta f(x, \theta_t)^T \nabla_\theta f(x', \theta_t). \tag{1}$$

To compare network kernels at different times, we use the kernel distance $S$ developed by [11]:

$$S(t_1, t_2) = 1 - \frac{\langle \mathcal{K}_{t_1}, \mathcal{K}_{t_2} \rangle}{\|\mathcal{K}_{t_1}\|_F \|\mathcal{K}_{t_2}\|_F} \tag{2}$$

This measures a normalized dissimilarity between two kernels. Since the NTK is positive semi-definite, the inner product is always non-negative between 0 (identical) and 1 (orthogonal). As it is a scale invariant distance, it quantifies the relative rotation between the linearized model at different training times. In practice, the dimension of the empirical kernel is large for multiple output networks as the ones we study here, and we use a subsample of $n = 256$ of training dataset to calculate the empirical NTK (which constitutes the empirical gradient directions) [17, 23]; higher $n$ or using test dataset to calculate the NTK gives a similar pair-wise distance score (Appendix Fig. 6).

We applied the kernel distance matrix to examine how the kernel changes over the course of optimization in our grokking setup. Surprisingly, the kernel changed substantially even before grokking (Fig. 1 b, right, red line highlighting the block structure transition to grok; Appendix Fig. 9). We observed similar kernel distance changes for the output scaling and weight decay schemes to induce grokking in MNIST (Appendix Fig. 7, Fig. 9). Based on prior literature, networks placed in the lazy regime via higher output scaling $\alpha$ or weight decay strategies might be expected to show minimal kernel distance before grokking [15, 24].

Our findings indicate that grokking networks for MNIST classification explore regions of weight space far from initialization before they start to generalize, indicating that learning prior to grokking is not lazy training. The change in the network function shows multiple stages of change before and after grokking, evident from the block structure of the kernel distance matrix that aligns with the slow performance increase before grokking and the steeper increase period after (Fig. 1 b right). During periods of significant changes in accuracy or loss trajectories, both the grokked and non-grokked networks exhibit a characteristic block-like pattern: a sudden increase in kernel distance (marking a substantial shift in the functional space) followed by relatively minimal movement in weight space while the performance trends persist. Since this pattern appears regardless of whether grokking occurs, it likely reflects a general property of neural network optimization rather than a mechanism specific to delayed generalization. As an alternative explanation, we hypothesize that grokking mirrors an improvement in the geometry of class representation, where the object manifold of different classes become more linearly separable which allows generic test data to be classified more robustly.

**Grokking is associated with a sharp change in network function space and manifold geometry.** We subsequently extended our investigation of grokking to the more complex EMNIST dataset [7], which encompasses a larger number of classes ($n_{\text{class}} = 47$, including numbers and digits) and presents a more challenging learning task. A critical aspect of rich learning, which we aim to demonstrate, extends beyond merely "escaping" the kernel learning regime; rather, it involves the network actively developing representations of task-related features and patterns within the data [2, 32]. The precise characterization of rich learning and its learned features is often task-specific; in the context of image classification, a prevailing hypothesis is that good object representation should be organized into linearly separable, disentangled class manifolds— a property widely sought after in both deep learning theory and cognitive science [10, 18]. Here we use the manifold capacity and geometry metrics over training, which provide a direct theory-based measure on the changes of representation geometry in a layer-wise manner.

We first provide a brief explanation of central terminology; more details can be found in [6, 8]. Given $N$-dimensional features in neural activation space represented by $P$ object manifolds, the manifold capacity $\alpha_C = P/N$ is defined by the critical number of object manifolds such that when $P < \alpha_C N$, manifolds can be shattered with high probability. In other words, when assigning random binary labels to the manifolds, they can be linearly classified most of the time. Large $\alpha_C$ implies well-separated manifolds in feature space. Two closely related properties of class object representation that describe the manifold geometry are reported here: $D_M$, the effective dimension and $R_M$, the effective radius. The two measures are defined by the anchor points of each class, which is a subset of training points that lies on the decision boundaries to other classes, and they describes the effective geometry of the class manifolds. The effective dimension captures the average dimension of the anchor points of given point cloud manifolds, while the effective radius measures the average norms of the anchor points that gives an estimate of the spread of the manifolds. A small $D_M$ implies that the anchor points of the point cloud occupy a low dimensional space, and a small $R_M$ implies that the anchor points of a given point cloud tightly group together, and both contribute to a bigger manifold capacity $\alpha_C$. As shown in Appendix Fig. 5, for a network trained with standard initialization on EMNIST, the manifold capacity metrics ($\alpha_M, R_M, D_M$) showed significant improvements as the network learned.

In our main result, we show the manifold geometry measures calculated at test activations (about 200 sample activations per class), as they are generic points that the network haven't seen but should classify. We also computed the measurement on both train and test and show that they track each other over training (see Appendix Fig. 13 and Fig. 14), thus providing similar information about the manifold geometry in this context.

When measured over the course of training in a grokking paradigm, we observed that the manifold capacity across layers $\alpha_C$ barely changed before grokking but increases steadily during grokking (Fig. 2 b) while in contrast the kernel has moved away from initialization. This association between generalization and a sharp change in manifold measures was observed in all three setups for induced grokking and across seeds (Appendix Fig. 15), although we also noticed variability to induce the delayed task improvement using different seeds for otherwise identical networks (Appendix Fig. 11). Manifold capacity measurements thus better correspond to the onset of delayed generalization than changes to the network tangent kernel (Fig. 2 c).

We chose not to focus on the loss metric, as we observed that with the AdamW optimizer, model performance can continue improving even as the loss increases (Fig. 2 a). This divergence between

loss and performance suggests that loss alone may not be a reliable predictor of classification performance nor the grokking behavior. While the increased loss in late training phase is not observed across all trained networks, we note great instability in the training loss curves with AdamW. Although our main results use AdamW in keeping with previous literature [15, 19], we also induced grokking using Adam with weight decay on the same network architecture (Appendix Fig. 12), where the loss exhibits a more conventional train-test gap pattern [12]. Notably, the manifold capacity across all layers shows stronger correlation with accuracy in the grokked network (Appendix Fig. 10).

The improved manifold capacity suggests that the weight space learns features that make object classes more separable, which implies more robust representations that enable the network to generalize on test data. This notion of changes in manifold geometry can be visualized in a UMAP projection (Fig. 2 d). In the early epochs, the object classes appear diffuse and poorly separated; as the network groks (around epoch 4000), the classes become increasingly distinct and tightly clustered, with clear boundaries between them. This visualization intuitively supports the insights gained from manifold capacity, that the network learns to extract features that effectively discriminate between object categories.

**Manifold geometry metrics reveal generalization property of the network function beyond test performance metric.** Now, we compare two networks that differ only by the output scaling parameter, one with large $\alpha = 0.5$ and delayed generalization training dynamics, and one with small $\alpha = 0.001$ and no delay (Fig. 3).

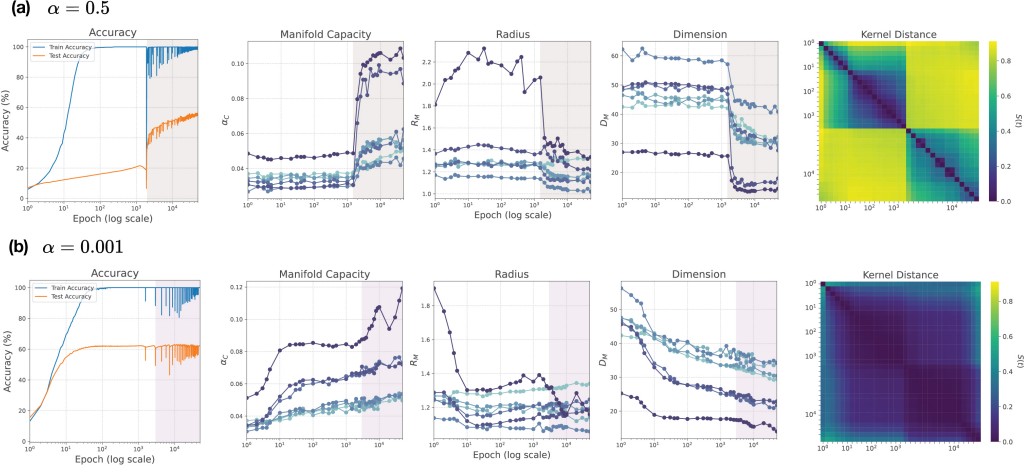

Figure 3: Training dynamics, manifold measures and kernel distance of two output scaling ($\alpha = 0.5, 0.001$). Networks were trained with $n = 2000$ samples. **(a)** The grokked network with output scaling parameter $\alpha = 0.5$. **(b)** The network with no delayed generalization with $\alpha = 0.001$.

With $\alpha = 0.5$, we observed a sharp increase manifold capacity $\alpha_C$ as for the early example network (Fig. 3 a). In line with our previous observations, the kernel continues to change both before and after grokking, but the manifold capacity remains low and stable until grokking occurs. In addition, changes in effective radius $R_M$ and effective dimension $D_M$ suggests transition to generalization is associated with both the representation becoming lower dimensional (decreased $D_M$), and the reduction of magnitude of the overall variance (decreased $R_M$). While all layers undergo similar transitions in manifold geometry, the effect is strongest for the deepest layers in the network.

For small $\alpha$, grokking is eliminated as the test performance closely tracks that of the training (Fig. 3 b). However, the capacity measurements reveal an interesting change in the network that happens in the late epochs where the performance becomes unstable. During this period (highlighted in light purple shade in Fig. 2), the capacity keeps increasing, and the decreased $R_M$ and $D_M$ suggest that the representation is drastically compressed. This "compression" has been shown to deteriorate the generalization ability on out-of-distribution data and can lead to higher catastrophic forgetting [21]. Moreover, the change is reminiscent of the memorization in late epochs revealed in network trained

in permuted labels, where manifold capacity keep increasing after the generalization phase, but start to over-fit on the permuted labels [30].

We thus suspected a slightly different type of memorization for the later epochs here: after the network has adapted to its internal representations, once over-trained, the network can start to discard some learned features and only rely on a small subset of features, which might be spurious or specific to a dataset.

**Feature learning can be gained and also lost.** Following the above rationale, we predicted that the "compression" of the representation's radius and dimension in the late stages of training, would decrease the robustness of generalization. This would be especially visible when evaluating data with slight distribution shifts from the training data.

To characterize how learned representations evolve through training, we evaluated the network's generalization capabilities under subtle distribution shifts using MNIST dataset. While both EMNIST and MNIST contain handwritten digits, their distributions differ due to EMNIST's additional letter classes and collection process variations. We assessed feature quality through linear probing: a linear classification layer was attached to the second-to-last layer and trained to classify MNIST digits (Fig. 4). This probes whether the learned features are general and transferable. For $\alpha = 0.5$, the linear probing shows improved performance around the grokking epoch, confirming the network's transition to meaningful feature learning. For $\alpha = 0.001$, an intriguing pattern emerges: while the probing accuracy shows strong performance during the early plateau of train/test accuracy, it drops sharply in later epochs when manifold capacity continues increasing. This suggests a form of representation collapse [26]: the network's features become increasingly specialized for EMNIST, potentially discarding general handwritten character features in favor of dataset-specific patterns. The sharp decline in linear probing performance indicates the features have become less general and transferable, consistent with overspecialization to the training data.

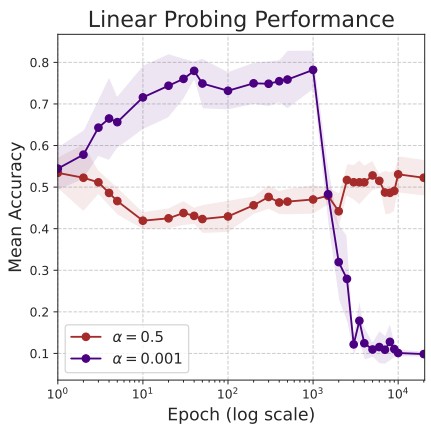

Figure 4: Performance under distribution shift. The mean accuracies on MNIST of the two networks from Fig. 3 across training epochs using linear probing. Shaded regions indicate standard deviation over 10 runs with randomly sampled MNIST data ($n = 1280$).

## 5 Discussion

Grokking constitutes an insightful phenomenon for investigating neural networks' generalization properties [13]. Prior theories attribute delayed generalization to a transition from lazy to rich learning regimes, suggesting networks remain close to initialization before grokking occurs. In this study, we examine these theoretical insights in the context of image classification, extending beyond the simpler tasks typically studied in the grokking literature. We test whether the theoretical insights from simpler settings generalize to more complex scenarios. We show that before grokking the over-fitting training performance is not necessarily achieved near initialization. but there still exists a sudden change in the network (indicated by the data-dependent NTK) when the network starts to grok, in line with Lyu et al. [20]'s theoretical prediction. To explain this change that is coupled with grokking, we examine several measures of the manifold geometry of layer representations. We find that the manifold geometry remains constant until the network groks, reminiscent of Davies et al. [9] and Varma et al. [31]'s findings. During the training, the network quickly learns heuristic patterns and over-fits on the inputs, then learns generalizable features, which is slow and can lag behind the training performance metrics.

We further investigate the performance under distribution shift for networks that do not achieve grokking. Our analysis reveals that features learned during early training can deteriorate in later epochs. While this degradation is not apparent when examining kernel changes alone, it becomes evident through capacity measurements. This phenomenon aligns with the late-epoch memorization

observed by Stephenson et al. [30] and the "misgrokking" concept introduced by Lyu et al. [20], where early-phase implicit bias leads to generalizable solutions, while late-phase bias results in overfitting. We propose that grokking should be viewed within a more comprehensive theoretical framework that captures the interplay between training biases and the transitions between overfitting and generalizing solutions, which will deeper insights into the dynamics of neural network learning.

Although we only focus on image classification in this work, we believe these insights could extend to other tasks. More importantly, we hope our results will motivate the community to evaluate future theoretical explanations of grokking on sufficiently complex datasets.

## Acknowledgments and Disclosure of Funding

This project originated from a class project in Prof. SueYeon Chung's course *Neural Network: Theory and Applications*, Spring 2024, where the first author became interested in representation learning and was introduced to manifold capacity theory. We thank Dr. Chi-Ning Chou for helpful discussions, and Burak M. Gürbüz for support throughout the project, and Dr. Sergey Shuvaev for revision advice. We thank the reviewers for their constructive suggestions.

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

# Appendix

## Additional tables & figures

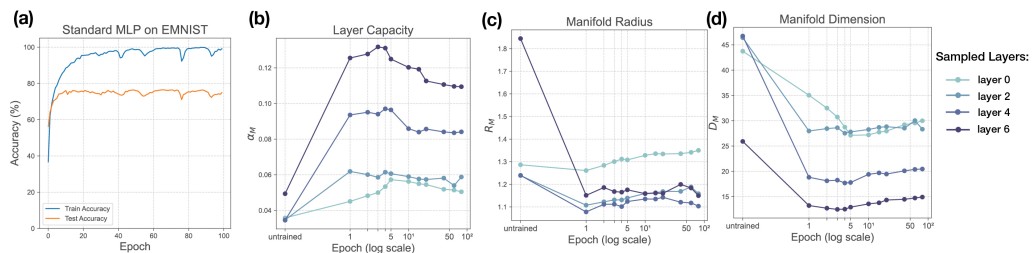

Figure 5: **Manifold capacity measurements on a standard initialized MLP network (no weight scaling, no output scaling, and trained with sufficiently large dataset).** Inspecting across layers, the later layers also show higher object manifold capacity as expected, since the later layers should learn high-level features that more directly contribute to classification [8].

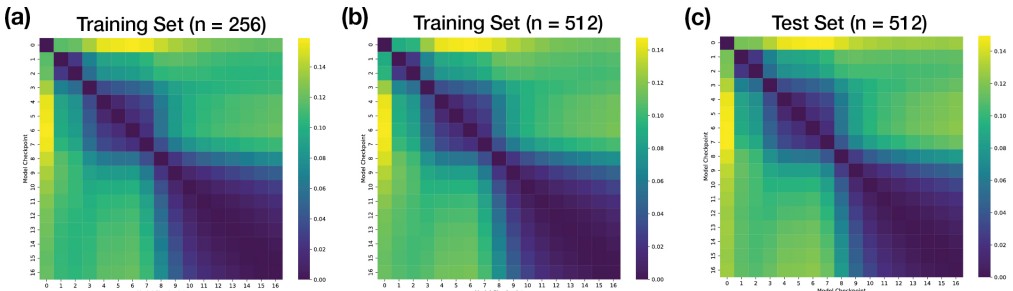

Figure 6: **Example pairwise kernel distance heatmap calculated at different samples.** The MNIST network with output scaling $\alpha = 0.5$ is shown here. **(a)** Using 256 training samples. **(b)** Using 512 training samples. **(c)** Using 256 test samples.

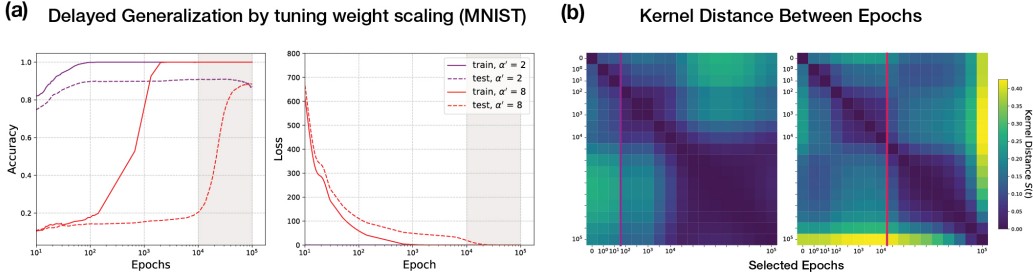

Figure 7: **Delayed generalization in MNIST by tuning weight scaling parameter** $\alpha'$. **(a)** Performance of example grokked and no-grok networks. **(b)** Kernel distance heatmaps. Left is for $\alpha' = 2$, right is $\alpha' = 8$.

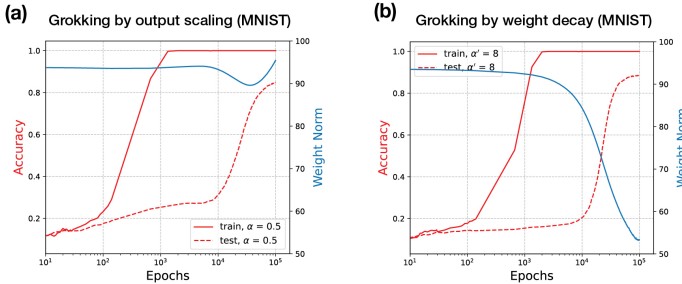

Figure 8: **Weight norm changes during grokking**. Weight norm decrease precedes improved test performance, but does not reflect all changes in the kernel (see Fig. 1), 7).

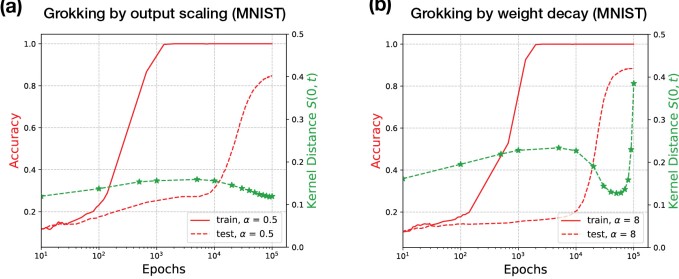

Figure 9: **Kernel Distance to network at initialization over the course of training.**

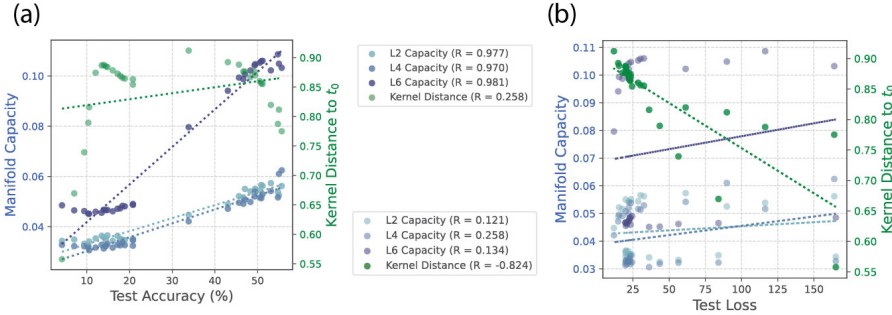

Figure 10: **Correlation between test performances and Manifold capacity measure and kernel distance to initialization.** **(a)** Correlation to test accuracy. **(b)** Correlation to test loss. We sampled manifold capacity of Layer 2, 4 and 6 for the visualization.

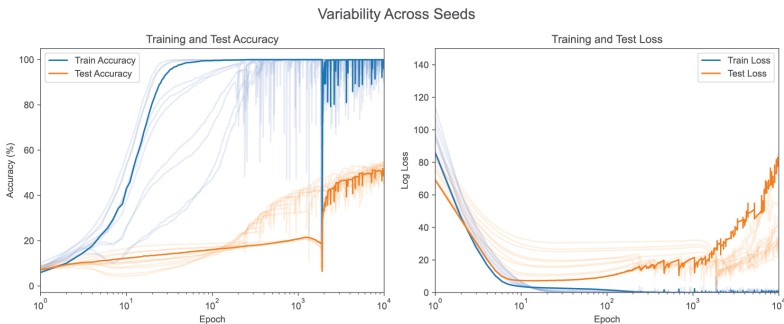

Figure 11: **Variability in induced grokking in EMNIST.** In this network, $\alpha = .5$, training samples $n = 2000$, with no weight decay.

| | sample size | weight scaling | output scaling | weight decay |
|---|---|---|---|---|
| **varying initialized weight norm** | 2000 | 10 | 1 | 0.001 |
| | 2000 | 5 | 1 | 0.001 |
| | 2000 | 1 | 1 | 0.001 |
| **varying sample size** | 1000 | 10 | 1 | 0.001 |
| | 2000 | 10 | 1 | 0.001 |
| | 5000 | 10 | 1 | 0.001 |
| **varying output scale** | 2000 | 10 | 0.5 | 0 |
| | 2000 | 10 | 0.1 | 0 |
| | 2000 | 10 | 0.001 | 0 |

Table 1: **Three setups to induce/eliminate grokking.**

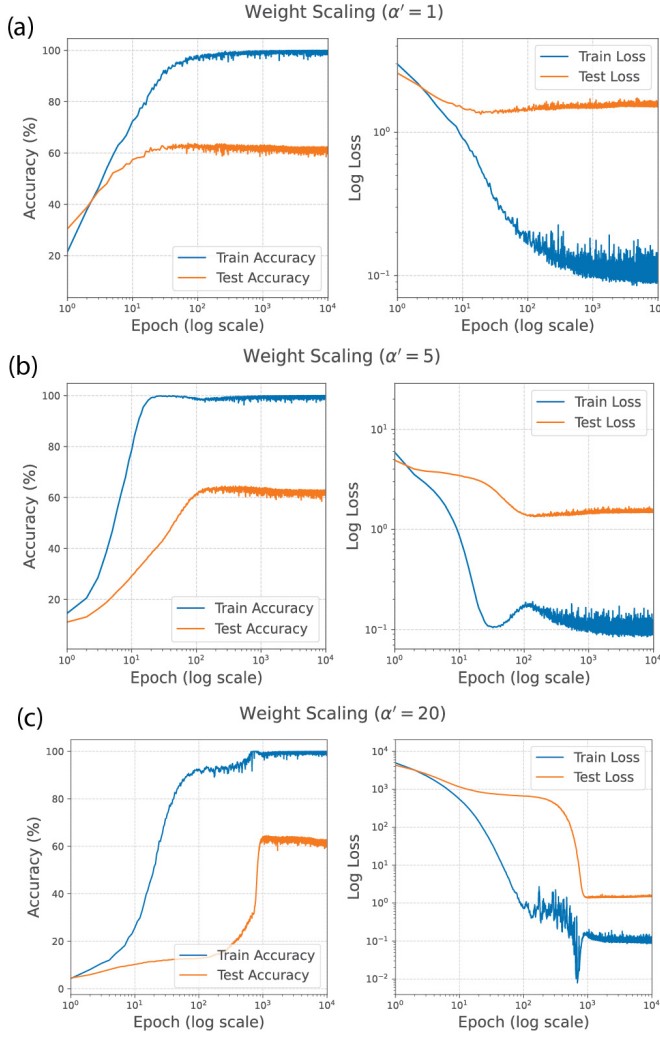

Figure 12: **Grokking induced with Adam optimizer with weight decay by tuning weight scaling parameter** $\alpha'$**.** We use the same model architecture as described in Section 3; and other setting the same with weight decay using AdamW optimizer (Table 1).

Figure 13: **Manifold capacity and geometry measures for both training and test set for the two networks in.** About 20 samples per class was used from training/ test dataset. The overall trends of changes are consistent for training and test samples.

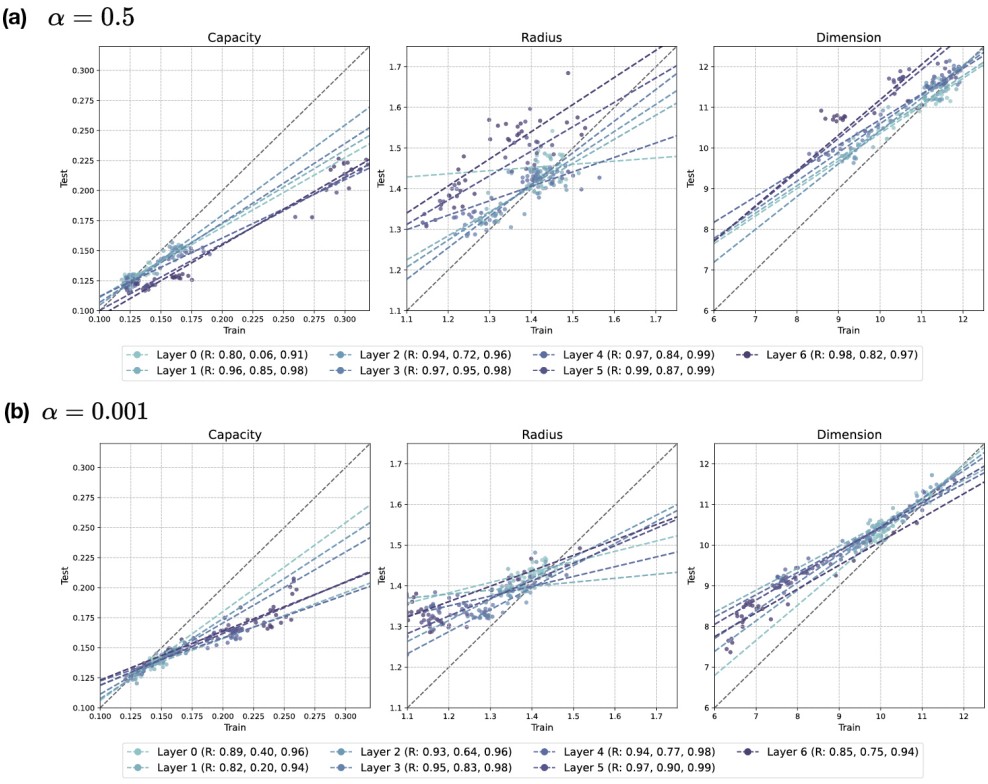

Figure 14: **Correlation of** $\alpha_C, R_M, D_M$ **between train samples and test samples.** Same two networks is shown as Fig. 13. The R-value (correlation coefficient) is shown in the legend, and overall correlation is high for all measures, and there is no big difference between the two different network setups.

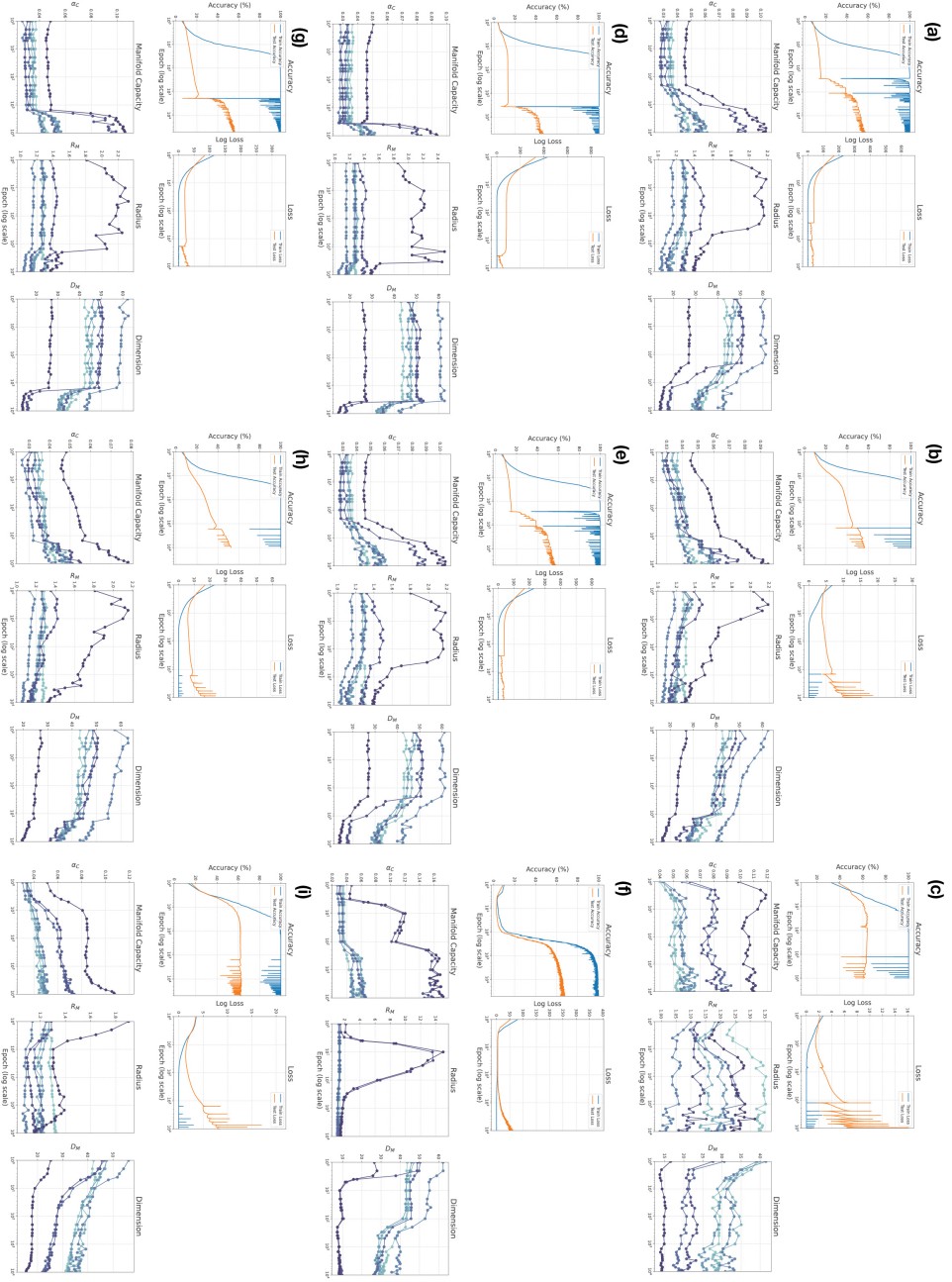

Figure 15: **Performance metrics (upper row) and manifold capacity and geometry measures (lower row), ordered the same as in Table 1. (a-c)**. Changed weight norm. **(d-f)**. Changed tranining set size. **(g-i)**. changed output scaling.

