# OpenReview forum: "Delays in generalization match delayed changes in representational geometry"
_NeurIPS.cc/2024/Workshop/UniReps — UniReps_

### Official Review · Reviewer_tnbr · 2024-10-05
**Review of submission #52**

**Rating:** 6
**Confidence:** 4

**Review:**

The authors study "grokking," where neural networks start generalizing only after overfitting for a long time. The authors find that changes in the Neural Tangent Kernel (NTK) happen before grokking, meaning rich learning starts earlier than expected. However, they show that the key factor in grokking is not NTK changes but a sudden improvement in how well the network separates data, measured by its representational geometry and manifold capacity.

Strengths:
- Although not proposing a completely novel idea, they build on existing theories (like those by Lyu et al.), and propose new ways to understand the grokking process through representational geometry and apply it to image classification tasks​. They provided thorough reasoning and explanation regarding behaviour on OOD data which I found interesting and important to include in study.
- Writing is clear for the most part.
- Nice visualization of plots showing manifold capacity during grokking.

Weaknesses:
- Lacking a more thorough explanation regarding importance of neural network training dynamics and effects on generalization and why grokking is important to study and how it is different from double descent would be useful to explain.
- Toy models were used. Would like to see how this behaves on larger datasets and bigger models.
-  The authors introduce manifold capacity and other geometry metrics as useful tools for analyzing generalization, I wanted to see if they can extend to some more justification as to why these metrics are the right ones to track grokking.

Question: Do you the authors plan to investigate long-term effects on larger epochs? They observe interesting behaviours related to "compression" and the risk of overfitting in later stages of training, but more thorough exploration of this could yield insights into preventing late-stage overfitting or misgrokking​?

---

### Official Review · Reviewer_Wgg2 · 2024-10-06
**Nice empirical study on grokking, NTK, and MMCR learning regimes for image classification**

**Rating:** 9
**Confidence:** 4

**Review:**

This paper is an empirical study on image classification that correlates the learning transition to grokking with geometric signatures from the NTK and MMCR. It starts by replicating recent theoretical results (Kumar et al., 2024) that posit grokking as a transition from lazy to rich learning regimes. Finding that the weight norm follows the transition to grokking as expected, the authors then find that sudden changes in NTK distance are a sufficient but not necessary condition for grokking: the NTK changes much earlier than the grokking point contrary to what is suggested in past literature. Finally, the authors propose that grokking coincides with meaningful changes in representation geometry, and find a temporally coupled phase transition in the manifold capacity to delayed generalization, for deeper network layers.

Overall, the paper is very well-written and easy to follow. There is a thorough engagement with previous theoretical results, the new empirical results are convincing, and the theme is aligned with the workshop. I would be in strong favor of accepting the paper.

---

### Official Review · Reviewer_iMZe · 2024-10-06
**Interesting method, but the core claim has not been demonstrated, among other issues**

**Rating:** 3
**Confidence:** 4

**Review:**

**Summary**
- This paper performs an empirical study of grokking on MNIST/EMNIST using a novel representational geometry method (”manifold capacity”). The paper proposes that manifold capacity offers improved explanatory power as a progress measure for grokking over parameter-based NTK methods popular in feature learning literature. In support of this proposal, the paper claims that whereas NTK changes precede the timing of generalization, manifold capacity coincides more closely with generalization.

**Strengths**
1. Proposing to study grokking and feature-learning from a representation lens rather than a parameter lens is an interesting and ambitious approach
2. The plots tracking manifold capacity, accompanied by UMAP visualizations (e.g. figure 2) are quite illuminating by showing how loss improves due to the rapid emergence of class-separability, and that manifold capacity offers a numerical measure to track this

**Weaknesses**
1. The paper’s core claim rests on the idea that “the NTK deviates from its initial state significantly before the onset of grokking, i.e., before test performance increases, suggesting that rich learning does occur before generalization”. This idea is not supported by the paper’s own figures.
  - In figure 1b, the core message is that the block-structures in the kernel distance matrix before the vertical colored lines suggest that the NTK changes significantly before the “onset of generalization”. This is supposed to demonstrate that the NTK and rich-lazy explanation (e.g. in ref 14) imperfectly track generalization in real-world task settings. This is untrue, as figure 1a shows both test loss and accuracy significantly improving before the vertical colored lines. It appears that NTK changes are coinciding with improvements in generalization throughout.
  - The authors might be better off using the typical modular arithmetic tasks which are more likely to demonstrate cleaner single-descent grokking behavior. It would be clearer to see if NTK changes happen before generalization in those settings.
  - In figure 1a, the loss for the small-alpha initialization is not visible at all due to scale. Loss behaviour is important to focus on when studying grokking since grokking in loss implies grokking in accuracy, but not the other way around (see ref 14 sections 3 and 16 for discussion).

2. The paper’s core claim is not demonstrated. The core claim is that changes in manifold capacity “better correspond to the onset of delayed generalization than changes to the neural tangent kernel (Fig. 2 c)”. There are several issues.
  - In figure 2c, the x and y-scale of the kernel distance matrix appear to be log-scale until 10^3, but then the scale arbitrarily changes, which prevents comparison with figs 2a-b. This prevents the reader from evaluating whether the claim - that manifold capacity tracks generalization improvement better than NTK change - is supported.
  - In figure 2a, it is apparent that the epochs shaded - which are intended to signify the “grokking” phase - are heuristically determined using accuracy/capacity rather than loss. As discussed, the choice to use accuracy rather than loss is at best arbitrary and at worst misleading. More importantly, since test loss undergoes both a first and second descent before the improvement in manifold capacity, the core claim does not appear to be supported.

3. The paper characterizes OOD (out-of-distribution) performance by training the network on EMNIST and performing linear probes on the representations of MNIST. Couple issues:
  - EMNIST contains handwritten digits and letters. MNIST contains just handwritten digits. Since the latter is essentially a subtask of the former I don’t think this qualifies as an OOD task.
  - The setup in Figure 4 is unclear and does not make much sense.
    - Why was a linear probe with limited (1000) samples used in each run? Since the MNIST labels exist as a subset of the EMNIST labels, why not just re-label the MNIST test set with 47-length logits and directly evaluate using the original readout layer?
    - What and how much data was used to train the linear probe? If it was only 1000 samples then linear probe performance may not be an accurate representation of the quality of the learned features

4. The paper’s core method - manifold capacity - is not adequately explained despite it not being well-known in feature learning literature, which prevents the reader from understanding the intuition and reasoning behind the work
  - It is not adequately explained how the “critical number of object manifolds where most manifold dichotomies can be shattered” is connected to the prevailing understanding of grokking. In section 5, the paper does attempt to explain manifold capacity in one paragraph, but this is quite haphazard. For example, they introduce the term “anchor points” without explaining what these are. The paper directs the reader to more details about the theory in refs 6 and 8, but these papers are general technical texts outside the context of feature-learning / generalization. The paper cannot be understood unless the core method is explained adequately within the same text, with a clear outline of why it might be informative to understand grokking

---

> ### Author Response · Authors · 2024-11-05
> **Author Response**
>
> We thank the reviewers for their thoughtful feedback and suggestions that have helped improve the clarity and rigor of our manuscript.
>
> Response:
> - We acknowledge that image classification exhibits a less distinct grokking pattern compared to modular arithmetic. We have clarified our interpretation of the kernel distance heatmap: while it tracks general changes in performance for both grokked and non-grokked networks, it does not specifically correlate with the grokking phenomenon or test accuracy improvements.
> - We agree that "distribution shift" is more appropriate than "OOD" in describing the MNIST evaluation. Our use of linear probing rather than direct classification allows us to specifically examine the learned representations in the feature space, which is more relevant for understanding how the network's internal representations evolve during training.
>
> Editing:
> - We have added justification for focusing on accuracy metrics rather than loss in this study.
> - We have improved the clarity of manifold capacity theory presentation and its key definitions.

---

### Official Review · Reviewer_eoet · 2024-10-07
**Grokking in image classification using Neural Tangent Kernels**

**Rating:** 7
**Confidence:** 4

**Review:**

The paper explores the intriguing phenomenon of grokking, where neural networks experience delayed generalization despite early overfitting, particularly in overparameterized models. It empirically investigates grokking in image classification tasks, focusing on the relationship between Neural Tangent Kernel (NTK) changes and representational geometry, with an emphasis on manifold capacity as a measure of generalization. The study is well-executed, using solid empirical methods and providing detailed analysis on EMNIST and MNIST datasets, successfully replicating earlier findings and expanding on them.

The study's introduction of manifold geometry metrics such as capacity, effective dimension, and radius, offers a novel lens to understand grokking beyond traditional NTK-based analysis. This extension of grokking theory to manifold geometry is a valuable contribution, and the paper demonstrates a strong grasp of representation learning and its connection to generalization. However, there are some questions about scalability and whether the insights extend to more complex datasets beyond MNIST/EMNIST.

---

### Decision · Program_Chairs · 2024-10-10

**Decision:**

Accept

**Comment:**

In light of the positive reviewers' feedback and relevancy of the submission, we are pleased to accept this paper for presentation at UniReps 2024. We kindly ask the authors to incorporate the reviewers' suggestions and feedback in the final camera-ready version of the manuscript.